# Performance Evaluation of Complex Equipment Considering Resume Information

**DOI:** 10.3390/e24121811

**Published:** 2022-12-12

**Authors:** Xiangyi Zhou, Zhijie Zhou, Guanyu Hu, Xiaoxia Han, Leiyu Chen

**Affiliations:** 1Missile Engineering Institute, PLA Rocket Force University of Engineering, Xi’an 710025, China; 2Institute of Computer and Information Security, Guilin University of Electronic Technology, Guilin 541000, China; 3Combat support college, PLA Rocket Force University of Engineering, Xi’an 710025, China

**Keywords:** performance evaluation, resume information, evidential reasoning, parameter calculation

## Abstract

It is of great significance to obtain the performance state of complex equipment to protect equipment and maintain its normal operation. The majority of the performance evaluation methods are based on test data, but resume information is not considered. With its wide applicability and completeness, the resume information can be used in the comprehensive evaluation of equipment in various non-testing situations. By incorporating resume information into the performance evaluation of complex equipment, the flexible use of test data and resume information can result in a more comprehensive and accurate evaluation. Therefore, this paper focuses on the evaluation method of complex equipment performance based on evidential reasoning (ER) considering resume information. In order to unify the test data and resume information in the same framework, a novel method is proposed to transform them into the ER-based performance evaluation. On this basis, according to the index types, different reliability calculation methods are put forward, with one being based on the first-order fitting coefficient of variation, and the other being based on average time to failure; the index weight is analyzed based on the method of expert weight construction. Then, the transformed information with reliability and weight are fused by the ER rule. Finally, a performance evaluation case of a certain inertial measurement unit (IMU) is conducted to verify the effectiveness of the proposed method.

## 1. Introduction

With the continuous improvement of the degree of industrialization, equipment is becoming more and more complex [1]. At present, complex equipment has penetrated into the activities of production and national defense construction, such as industrial equipment, rockets, and so on [2]. An equipment system is a special kind of complex giant system that has the characteristics of a complex structure and high information coupling degree [3]. The structure, function and interaction of complex equipment are highly integrated and coordinated, showing the characteristics of a typical complex system, which is the root cause of the complexity of the complex equipment system [4]. In order to maintain good performance operation and continuously and efficiently provide convenience for people, it is necessary to check the equipment itself and evaluate its performance status [5]. On the one hand, complex equipment systems are related to people’s livelihood and national defense, and once an accident occurs, it poses a huge threat to people’s lives and property. A performance evaluation can find the hidden dangers of equipment systems in time and put forward countermeasures, such as reducing workload, improving working standards or decommissioning and scrapping to reduce losses. On the other hand, complex equipment systems are expensive and it is more economical to check before the accident than to maintain after the accident [6]; moreover, regular maintenance and replacement without performance evaluation will also cause huge waste. A performance evaluation can provide the basis for maintenance, and accurate maintenance can save resources to a greater extent [7]. However, due to the increasing complexity of equipment, it is challenging to evaluate the performance of it [8].

There are two kinds of information—i.e., quantitative data and qualitative information—that can be used for equipment performance evaluation [9]. Quantitative data refer to the information that exists in the form of a quantity and can be measured. Generally, it refers to test information. Qualitative information refers to the non-quantitative means that are used to express the nature of the equipment, including mechanism information, resume information, etc. [10]; however, the numerical value in the resume information is essentially a kind of qualitative knowledge. Since it cannot be directly related to performance, it needs a certain transformation process. Among them, the equipment resume information is mainly used to establish the equipment quality file information. By establishing the file information of each piece of equipment, the initial configuration of each unit before leaving the factory and the latest configuration after leaving the factory can be queried so as to trace the resume information, such as equipment configuration, replacement records in field use, spare parts quantity, etc., and provide the basis for an equipment performance evaluation [11]. For example, during the consultation in the hospital, doctors should not only check the examination results, but also check the medical records and ask for medical history so that doctors can draw exact conclusions and patients can get accurate and effective treatment. Each specific examination item is a piece of quantitative test information, and medical records are qualitative resume information. If the diagnosis and treatment plan is not based on the medical records, the patient may not only be prevented from healing, but this may also cause secondary damage to the body. It is the same with an equipment performance evaluation, which requires both “inquiry” and “palpation”. This means that it is necessary to check the resume and test to accurately grasp its performance status.

However, at present, most of the research is based on test data without fully utilizing the resume information. The test data can intuitively characterize the performance characteristics. For example, Dong et al. introduced zero bias, zero drift and scale factor as the evaluation indices of laser IMU [12]. The final evaluation result was obtained by weighting and fusing all indices. Chao et al. used a neural network-based method to evaluate the performance of transformer bushing after an earthquake [13]. The aforementioned studies are based on the complete reliability of quantitative test data. Nevertheless, the performance changes caused by the external environment, self-wear, etc. are accidental, and there are systematic errors in measurement. Hence, the equipment performance cannot be fully reflected by the test data, especially the implicit and abrupt information in the test data that cannot be represented in a limited number of tests. The evaluation based on quantitative data has low requirements for the mechanism analysis of the system, and the evaluation model can be established by using statistical data; however, it requires a high accuracy of data collection and relies on large sample information, thereby not making full use of mechanistic information. Essentially, this belongs to the “black-box” model, which is not complete. The evaluation of qualitative information represented by resume information is not affected by observation data. The evaluation model based on equipment performance is directly constructed through mechanism analysis and expert knowledge, which has strong transparency. Therefore, there is an urgent need for a semi-quantitative evaluation method that can handle both quantitative information and qualitative information.

The evaluation method based on semi-quantitative information can effectively solve the problem of the structural safety evaluation of complex systems with small samples [14]. Through information fusion, applying them to the performance evaluation of complex equipment can effectively solve the problems of incomplete test data and unspecific resume information in the current performance evaluation of complex equipment systems. Commonly used information fusion methods include subjective Bayes, fuzzy set theory, evidential reasoning (ER) rule and so on. The Bayesian model-based method can result in the posterior probability of knowledge being true when the prior probability of knowledge being true and the conditional probability observed from the data source are both known. In the equipment performance evaluation, the current performance evaluation result of the equipment is the conditional probability, so the prior probability of the equipment needs to be obtained through several full life cycle experiments before leaving the factory, which is very difficult to satisfy. Since the time cycle is long and the cost is high, a lot of time and manpower and material resources will be spent. The method based on fuzzy set theory takes the object to be investigated and the fuzzy concept reflecting it as a certain fuzzy set. Then, an appropriate membership function is established. Through the operation and transformation of fuzzy sets, the fuzzy objects are analyzed. Based on fuzzy mathematics, fuzzy set theory studies imprecise phenomena, but fuzzy sets cannot describe all fuzzy phenomena correctly, which is one of the reasons that limit the application of fuzzy sets [15]. Similarly, the complexity of complex equipment lies in the inability to clearly describe the coupling relationship and evolution law of a complex structure. Therefore, the engineering practice of complex equipment has limited the application of fuzzy set theory methods. Among the information fusion methods, the ER rule is a method of fusing multiple information based on D-S evidence theory and decision theory. The ER rule can make full use of qualitative or quantitative information, such as expert experience and experimental data, to quantitatively evaluate the evaluation results. The ER method describes all kinds of uncertainties in multi-attribute decision-making problems by establishing a unified belief degree framework, and quantitative information and qualitative information can be processed and fused under this framework so as to obtain the evaluation results [16]. The ER rules can be applied to the complex equipment performance evaluation considering the resume information in this paper. As a general extension of the probabilistic reasoning mode, the ER rule covers Bayesian reasoning. Traditional Bayesian reasoning methods can be regarded as a special case of the ER rule [16], so the ER has a broader application space. At the same time, the evaluation level in the identification framework of ER has clear concepts and is independent of each other. Uncertainty is inherited and transmitted under this framework so that the uncertainty factors can be effectively expressed, which is more practical than the fuzzy set theory [7].

In view of the above advantages of the ER rule in equipment performance evaluation, a complex equipment system performance evaluation method based on the ER method is proposed for the equipment performance evaluation problem considering resume information. It innovatively integrates resume information into the complex equipment performance evaluation index system and proposes a weight and reliability calculation method suitable for resume information and test data in the information fusion method based on ER. Finally, the effectiveness of this method is verified in the experiment of IMU.

The remainder of the article is organized as follows. In Section 2, the problems faced by the performance evaluation of complex equipment are sorted out and described. Then, the method to solve the problem is put forward in Section 3, and the input transformation model, weight and reliability calculation method are explored. In Section 4, a case study is provided to illustrate the feasibility and effectiveness of the equipment performance evaluation considering resume information in an engineering practice. The conclusion of this article is presented in Section 5.

## 2. Problem Formulation

### 2.1. Related Work

As mentioned in the introduction, the authors usually only consider the test data, but not the resume information, in the performance evaluation of complex equipment. The latest work is the practice and innovation of complex equipment in various industries.

In the performance evaluation of wireless sensor networks, Cheng et al. took the positioning accuracy as the evaluation objective, and took quantitative information, anchor position error, IMU error and TOA error as indicators, and then verified that the influence of anchor position error on the positioning results could not be ignored [17]. This provides a new basis for the benchmark of the IMU/TOA fusion positioning system and the reference lower bound for improving the performance of the positioning algorithm. An electric pump is a typical kind of complex equipment that is widely used in dams, wells, rivers, oceans and other drainage needs [18]. The performance of the electric pump will be jointly determined by the diameter of the drum, length of the drum (number of coils), RPM of the drum, water speed, tube diameter and depth of immersion, and finally, the performance will be measured in the form of output head and pump displacement. In the field of clean energy, a photovoltaic grid charging station based on a battery energy storage system is also a complex equipment system [19]. Its performance evaluation takes economic cost, environmental pollution and energy consumption as quantitative indicators. Through the evaluation results, the threshold of power, the optimal scheme is obtained by comparison. Neda Neisi studied and provided a method of modeling and analyzing complex systems without prior information [20]. Taking rotating machines as an example, the rotor supported by four rolling element bearings (REBs) was studied numerically and experimentally.

The above advanced research is based on the performance evaluation of test parameters or test data, and the resume information is not considered in the index system. Resume information is widely used in equipment management. Traditional performance evaluation methods do not make use of this, which on the one hand leads to the waste of information resources, and on the other hand leads to inaccurate evaluation results. The following research on equipment design and management considers resume information, but there is no quantitative test data involved in the performance evaluation and the testability of the test data itself is still taken as an index. The performance evaluation method in light of the qualitative knowledge is mainly based on FMEA (failure mode and effects analysis) [21], fault tree analysis [22], finite element analysis [23], Petri net [24] and AHP (analytic hierarchy process).

By analyzing the multi-sensor monitoring data of various parts of self-propelled artillery systems, Wang calculated the characteristic parameters of multi-sensors and obtained the combined weights of each characteristic parameter and evaluated the health index of multi-characteristic parameters of equipment, thus realizing the classification of the health index of characteristic parameters of complex equipment [25]. An et al. put forward that reliability, maintainability, supportability, safety, testability and environmental adaptability should be taken as performance evaluation indices, and the evaluation results can be obtained by evidence reasoning with expert empowerment [26]. The above indicators are qualitative information, all of which are not convincing through the way of expert empowerment, but they still have certain reference significance.

### 2.2. Problem Formulation

Whether it is a single weapons system or a combined complex equipment system, each system and piece of equipment does not exist in isolation, but rather has a specific coupling relationship with other systems and equipment [27]. This multilateral connection relationship is the physical basis of why the equipment system becomes a complex system [28]. The collection of all these relationships in the equipment system constitutes the structure of the complex equipment system, and it directly describes the existing form of the complex equipment system [29]. For example, the rocket control system is composed of various subsystems, such as the flight control system, attitude control system and safety self-destruction system, and the performance indices of the subsystems are different. The performance of each subsystem constitutes the performance index of the rocket, and each subsystem is composed of each module, such as the attitude control system consisting of nozzle, on-board computer and navigation system [30]. Based on this, the index evaluation system of a complex equipment system needs to be layered according to the characteristics of the system. The process is as follows.

In Figure 1, $\left(u_{1},\cdots ,u_{n},u_{n+1},\cdots ,u_{L}\right)$ represents input information, $\left(u_{1},\cdots ,u_{n}\right)$ indicates the input of resume information, $\left(u_{n+1},\cdots ,u_{L}\right)$ indicates test data input, $\left(e_{1},\cdots ,e_{n},e_{n+1},\cdots ,e_{L}\right)$ represents the transformation of input information to the first layer of evidence, $\left(e_{1},\cdots ,e_{n}\right)$ indicate resume evidence, $\left(e_{n+1},\cdots ,e_{L}\right)$ indicates test evidence, $\left(\omega_{1},\cdots ,\omega_{n},\omega_{n+1},\cdots ,\omega_{L}\right)$ indicates the relative importance of the evidence, $\left(r_{1},\cdots ,r_{n},r_{n+1},\cdots ,r_{L}\right)$ indicates the reliability of the source of evidence related to the result, $e_{s}$ represents the static evidence of equipment, $e_{d}$ indicates the dynamic evidence of equipment, and $\left\{\left(D_{1},\beta_{1}),(D_{2},\beta_{2}),\cdots ,(D_{m},\beta_{m}\right)\right\}$ indicates the fusion result.

In the evaluation system, first of all, the input of underlying indices needs to be transformed into the general form of evidence, and the weight $r_{i}$ and reliability $w_{i}$ of the indices in the transformation process should be retained. Then, in the first layer of fusion, according to the source of evidence, the evidence is layered and fused (F−Ⅰ,F−Ⅱ) into equipment static evidence $e_{s}$ or equipment dynamic evidence $e_{d}$. This process can be done single or multiple times. Finally, the static evidence of equipment and the dynamic evidence of equipment are fused (F−Ⅲ) to get the evaluation result $\left\{\left(H_{1},\beta_{1}),(H_{2},\beta_{2}),\cdots ,(H_{m},\beta_{m}\right)\right\}$.

The following problems are faced in this evaluation process:

**Problem** **1.**
*First of all, in the process of transforming the input*

$\left(u_{1},\cdots ,u_{n},u_{n+1},\cdots ,u_{L}\right)$

*of the underlying indices into the general form of evidence*

$\left(e_{1},\cdots ,e_{n},e_{n+1},\cdots ,e_{L}\right)$

*, the indices are multi-sourced. There are two main sources of information: (1) Resume information, which indirectly reflects the working state of equipment by recording completed behaviors and tasks; as the resume information remains unchanged for a long period of time, it belongs to static information. (2) Test data, which directly or indirectly reflect the working state of equipment through the test equipment, given that test data often reflect the current state of equipment, and change with time, which belongs to dynamic information. Both of them reflect the working state of the equipment. The former is quantitative information, such as voltage, current, speed, etc., while the latter is qualitative information, such as times of installation and inspection, transportation, etc. The character of the two kinds of information is quite different. How to fuse resume information and test data in a unified framework is one of the current difficulties.*


**Problem** **2.**
*The indices of complex equipment systems are various and their importance and influence on the results are different. The reliability of knowledge sources is also different due to the way of acquiring knowledge, and the weight and reliability of test data and resume information obtained in different environments are also different. When using ER to fuse information, the weight*

$w_{i}$

*and reliability*

$r_{i}$

*of each piece of evidence*

$e_{i}(i=1,2,\cdots ,L)$

*should be taken into account. The weight*

ω

*is the relative importance among the evidence, and the reliability*

r

*is the inherent characteristic of judging the results correctly, which is the key part of the description of the evidence. There are many sources of research objects in this paper, and the deviation of weight and reliability will bring great deviation to the evaluation results and also bring difficulties to the determination of weight and reliability calculation. How to correctly measure and reasonably use weight and reliability is another problem that we are currently facing.*


## 3. Evaluation Method of Equipment Performance Based on ER Rules

Based on the above two questions mentioned in Section 2, this section aims to establish a performance evaluation model for a complex equipment system considering the resume information. Firstly, the input information is converted into the form of belief degree, then the weight and belief degree are determined according to the input type. Last, the obtained information is fused by ER rules to get the performance evaluation results.

### 3.1. Input Information Transformation

The input information of a complex equipment system is various, such as voltage, oil pressure, service life, maintenance, etc. Each index directly or indirectly reflects the performance state of the equipment, but the information of each index is not uniform. For example, the lower the purity of the oil in the hydraulic system, the higher the possibility of its performance deterioration; if the bias of the navigation inertial unit is too large or too small, the possibility for equipment performance deterioration is higher. Therefore, it is necessary to describe the index information in mathematical language and convert the index information into the form of belief degree, so that the fusion and condition evaluation can be carried out, which lays the foundation for the subsequent formulation of a maintenance strategy and situation prediction.

The ER rules can effectively describe all kinds of uncertain inputs and realize the quantitative expression of evidence. Given an input, it can be equivalently transformed into the following belief distribution form:(1)$$S(x_{i})=\{(H_{n},\beta_{n,i}),n=1,2,\cdots ,N;i=1,2,\cdots ,L\}$$

Among them, $x_{i}$ can be both qualitative knowledge and quantitative information. $\left(H_{n},\beta_{n,i}\right)$ indicates that the i-th input is evaluated as level $H_{n}$ with belief degree of $\beta_{n,i}$, N indicates the number of evaluation levels, and L indicates the number of input information. The methods of input transformation of complex equipment in different forms are given below [31].

(1) Quantitative input transformation technique based on test information.

If the input quantitative information is $x_{i}$, the corresponding reference value is $h_{i,j}$
(i=1,⋯,L,j=1,⋯,J), where the number of reference values J is indicated. At this point, the decision maker or expert can establish a mapping relationship between the numerical value $x_{i,j}$ of $x_{i}$ and its reference value $h_{i,j}$, namely:(2)$$x_{i,j}\;\;means\;\;h_{i,j}$$

The formula of $\beta_{i,j}$ is as follows:(3)$$\left\{\begin{array}{l} \beta_{i,j}=\frac{h_{i,j+1}-x_{i,j}}{h_{i,j+1}-h_{i,j}},\;h_{i,j}\leq x_{i,j}\leq h_{i,j+1},\;j=1,\cdots ,J-1\\ \beta_{i,j+1}=1-\beta_{i,j},\;h_{i,j}\leq x_{i,j}\leq h_{i,j+1},\;j=1,\cdots ,J-1\\ \beta_{i,k}=0,\;k=1,\cdots ,J,\;k\neq j,j+1 \end{array}\right.$$

(2) Qualitative input transformation technique based on resume.

The input information of this transformation technique is qualitative, and the transformation technique is similar to the quantitative input transformation technique. The main difference between these two kinds of information is that the resume information needs to be further described before the transformation to make it conform to the general evidence expression form.

Generally, when the equipment leaves the factory, it will be equipped with a product certificate and management personnel to record the resume information. The indices vary from equipment to equipment, and a model is established according to the characteristics of the indices:(4)$$y=\frac{A_{2}-(\alpha_{1}\cdot x_{1}+\alpha_{2}\cdot x_{2}+\cdots +\alpha_{n}\cdot x_{n})}{A_{2}-A_{1}}$$

In the above formula, $x_{1}$, $x_{2}$, $\cdots ,x_{n}$ is the qualitative index level of equipment; $\alpha_{1}$, $\alpha_{2}$, $\cdots ,\alpha_{n}$ and $\alpha_{i}$ are the corresponding coefficients of $x_{i}$, which can be obtained by consulting the national and industry standards; $A_{2}$ is the maximum value or theoretical limit value of its single index; $A_{1}$ is its minimum value; and y is the transformed value of the quantitative expression of qualitative knowledge.

Taking the transformation of maintenance information as an example, according to the degree of failure, the maintenance situation can be divided into three levels: minor repair, medium repair and major repair. A model can be established by combining the maintenance times and levels:(5)$$y=\frac{A_{2}-(\alpha_{1}\cdot x_{1}+\alpha_{2}\cdot x_{2}+\alpha_{3}\cdot x_{3})}{A_{2}-A_{1}}$$

In the above formula, $\alpha_{1}$, $\alpha_{2}$ and $\alpha_{3}$ are the failure coefficient corresponding to minor repair, medium repair and overhaul, respectively; $x_{1}$, $x_{2}$ and $x_{3}$ are the number of minor repairs, medium repairs and major repairs in the record; $A_{2}$ is the maximum value or theoretical limit value of its maintenance transformation value; and $A_{1}$ is the minimum value of its maintenance transformation.

As shown in Table 1, the equivalent transformation coefficient of equipment maintenance is obtained by referring to GJB 6288-2008. Among the equivalent transformation coefficients of medium and minor repairs, it can be determined according to the specific maintenance situation of individual equipment. For example, the equivalent coefficient is different between changing a screw and changing a chip [32].

For the convenience of research, the median of the equivalent transformation coefficient, $\alpha_{3}=1$, $\alpha_{2}=0.4$, $\alpha_{1}=0.045$ was used. By assuming that major repairs being needed twice, medium repairs being needed 4 times and minor repairs being needed 10 times are the maximum maintenance value of this single machine, and by taking the maximum value of the maintenance transformation value as $A_{2}=4.05$, y=0, a minimum value of maintenance transformation is $A_{1}=0$, y=1
$y_{1}$ of the maintenance situation is obtained.
(6)$$y_{1}=\frac{4.05-(0.045\times x_{1}+0.4\times x_{2}+x_{3})}{4.05}$$

In fact, the input information transformation technique is not limited to Formulas (3) and (5) but depends on the relationship between the input and output in the decision scene of complex equipment. The calculation expression can be linear (such as trigonometric function, trapezoidal function, etc.) or nonlinear (such as exponential function, logarithmic function, compound function, etc.).

### 3.2. Determination of Weight and Reliability

In information fusion, the importance of an information source is expressed by the weight given to the information source by the fusion system designer, which is relative to other information sources. The reliability of an information source indicates its ability to provide a correct assessment or solution to a given problem.

(1) Weight calculation

The weight of evidence is the relative importance of this evidence compared with other evidence. According to the contrast object, the weight calculation is different; however, homologous evidence means that all the evidence comes from the same experiment, but heterogeneous evidence is the opposite. Contrast objects are divided into homologous and heterogeneous categories.

A. Homologous evidence: the weight of evidence composed of homologous data measured under the same conditions is 1.
(7)$$w_{1}:w_{2}:\cdots :w_{L}=1:1:\cdots :1$$

B. Heterogeneous evidence: For different conditions, different test equipment, different operators and different time periods, the weight of evidence is different. For example, in the annual maintenance of equipment, the measured IMU data has a long-time span, so the above method cannot be used to determine the weight, and the method of expert direct weight construction can be used to assign the weight.

Expert direct empowerment requires expert experience. Direct weighting is a weighting method that is directly assigned to each evaluation index according to the intuitive judgment of the weighting person and measured by the degree of importance. When the weights are assigned, the proportional method is usually adopted—the weight ratio $\left(p_{1}:p_{2}:\cdots :p_{L}\right)$ of L weighted objects.

Then, the relative number of proportions, i.e., the specific weight, is calculated.
(8)$$\omega_{j}=p_{j}/\sum p_{j}(j=1,2,\cdots ,L)$$

This method is usually used when experts have rich experience and few evaluation indices. Similar methods to deal with expert weight include analytic hierarchy process, the Delphi method, etc.

(2) Reliability calculation

The reliability of evidence is the inherent characteristic of evidence as a correct judgment of results. Similarly, different sources of evidence have different methods of calculating the reliability of evidence.

A. Reliability calculation based on test data

For the equipment, due to the interference of itself and the outside world in the testing process, the observation data of monitoring indices show different fluctuation rules at different times. This fluctuation may be caused by environmental changes, such as vibration, sound waves, temperature and humidity changes, etc., or it may be system noise. This change is random, which can reflect the reliability of this index to a certain extent; the greater the fluctuation, the lower the reliability. On the contrary, the smaller the fluctuation is, the higher the reliability is. In the practice of a multi-index comprehensive evaluation, this may happen: the data tend to show a certain trend; for example, with the accumulation of start-up time, the temperature of the equipment testing environment rises, the measurement resistance increases, and the current decreases. This trend change cannot be equated with fluctuation, so it is necessary to describe the change trend of data, which can be obtained by least square fitting. In view of this, the variation-based weighting coefficient (CVBW) and least square fitting can be used to determine the index reliability according to the fluctuation law of data.

The coefficient of variation formula of each index of coefficient of variation method is as follows:(9)$$V_{i}=\sigma_{i}/{\bar{x}}_{i},i=1,2,\cdots ,n$$

In the formula, $V_{i}$ is the coefficient of variation of the i index, also known as the coefficient of standard deviation; $\sigma_{i}$ is the standard deviation of the i index; and ${\bar{x}}_{i}$ is the average of index i.

The reliability of each index is:(10)$$r_{i}=V_{i}/\sum_{i=1}^{n}V_{i}$$

When there is a functional relationship $r_{i}=f(x_{1},x_{2},\cdots ,x_{n})$ between the reliability $r_{i}$ of an index and the single evaluation value $x_{i}$ of the index, the least square method is used to minimize the sum of squares of errors between these calculated data and the actual data.
(11)$$\hat{\theta }=\arg\min(J(\theta ))=\arg\min\{\sum_{i=1}^{m}{\left(f_{\theta }(i)-x_{i}\right)}^{2}\}$$

$f_{\theta }(i)$ is the fitted function, then the weight obtained by the coefficient of variation method after fitting by the least square method is as follows.
(12)$$r_{i}^{'}=V_{i}^{'}/\sum_{i=1}^{n}V_{i}^{'},V_{i}^{'}=\sqrt{{\left(x_{i}-f_{\theta }(i)\right)}^{2}}/f_{\theta }(i),i=1,2,\cdots ,n$$

B. Reliability calculation based on resume information.

For the resume information, there is only single evidence, and there is no data fluctuation. Obviously, the reliability calculation method based on the coefficient of variation and the least square method cannot be adopted. In system engineering, reliability is the probability that a product can complete a specified function under a specified condition and within a specified time. The record of complex equipment also comes from maintaining the functional operation of equipment, so the reliability calculation method based on life can be adopted [33].

The product life t is a random variable, where the reliability is $r\left(t\right)=p\left(T>t\right)$ and t is the current duration. The T-time reliability refers to the probability that the product will complete the specified function within [0,t]. The unreliability is $f\left(t\right)=P\left(T<=t\right)$, the unreliability at time t, which indicates the probability of product failure within [0,t]. Obviously, $r\left(t\right)+f\left(t\right)=1$.

The average life span is the average of life span. For irreparable products, it refers to the average working time before the product fails, which is usually recorded as MTTF (Mean Time to Failure). In the process of testing before the product leaves the factory, the measured life data is $t_{1},t_{2},\cdots ,t_{N_{0}}$. According to the definition, the estimated average working time before failure is $MTTF=(\sum t_{i})/N_{0}$. When the life is a continuous random variable, the average working time before failure is as follows:(13)$$MTTF=\int_{0}^{\infty }R(t)dt$$

Then, when the average working time before failure can approximate the effective service time, the reliability is as follows:(14)r(t)=t/MTTF

### 3.3. Information Fusion Based on ER Rules

Assuming that the identification framework is $\Theta =\{H_{1},\cdots ,H_{N}\}$, the evidence $e_{i}$ can be expressed as the belief distribution form shown in Formula (1), and the weight and reliability of $e_{i}$ are $w_{i}$ and $r_{i}$; then, the weighted belief distribution of evidence with reliability is defined as:(15)$$m_{i}=\{(\theta ,{\tilde{m}}_{\theta ,i}),\forall \theta \subseteq \Theta \;;\;(P(\Theta ),{\tilde{m}}_{P(\Theta ),i})\}$$

Among them,
(16)$${\tilde{m}}_{\theta ,i}=\left\{\begin{array}{l} 0\;\;\;\;\;\;\;,\;\theta =\emptyset \\ c_{rw,i}m_{\theta ,i}\;\;\;\;,\;\theta \subseteq \Theta ,\theta \neq \emptyset \\ c_{rw,i}(1-r_{i})\;\;,\;\theta =P(\Theta ) \end{array}\right.$$

In Formula (15), $c_{rw,i}=1/\left(1+w_{i}-r_{i}\right)$ represents a normalization coefficient, so that $\sum_{\theta \subseteq \Theta }{\tilde{m}}_{\theta ,i}+{\tilde{m}}_{P(\Theta ),i}=1$. For any two independent pieces of evidence $e_{i}$ and $e_{j}$, assuming that their belief distribution can be expressed by Formula (16), the joint support $p_{\theta ,e(2)}$ of $e_{i}$ and $e_{j}$ for proposition θ is determined by the following formula:(17)$$p_{\theta ,e(2)}=\left\{\begin{array}{l} 0\;\;\;\;\;,\;\theta =\emptyset \\ \frac{{\hat{m}}_{\theta ,e(2)}}{\sum_{A\subseteq \Theta }{\hat{m}}_{A,e(2)}}\;\;\;,\;\theta \subseteq \Theta ,\theta \neq \emptyset \end{array}\right.$$
(18)$${\hat{m}}_{\theta ,e(2)}=[(1-r_{i})m_{\theta ,j}+(1-r_{j})m_{\theta ,i}]+\sum_{A\cap B=\theta }m_{A,i}m_{B,j},\forall \theta \subseteq \Theta$$

Generally, for L piece of independent evidence $E=\{e_{1},e_{2},\cdots ,e_{L}\}$, their joint support $p_{\theta ,e(L)}$ for proposition θ can be obtained by continuously iterating the following formula:(19)$${\hat{m}}_{\theta ,e(k)}=[(1-r_{k})m_{\theta ,e(k-1)}+m_{P(\Theta ),e(k-1)}m_{\theta ,k}]+\sum_{A\cap B=\theta }m_{A,e(k-1)}m_{B,k},\forall \theta \subseteq \Theta$$
(20)$${\hat{m}}_{P(\Theta ),e(k)}=(1-r_{k})m_{P(\Theta ),e(k-1)}$$
(21)$$m_{\theta ,e(k)}=\left\{\begin{array}{l} 0\;\;\;\;\;\;\;,\;\theta =\emptyset \\ \frac{{\hat{m}}_{\theta ,e(k)}}{\sum_{A\subseteq \Theta }{\hat{m}}_{A,e(k)}+{\hat{m}}_{P(\Theta ),e(k)}}\;\;,\theta \neq \emptyset \end{array}\right.p_{\theta ,e(k)}=\left\{\begin{array}{l} 0\;\;\;\;,\;\theta =\emptyset \\ \frac{{\hat{m}}_{\theta ,e(k)}}{\sum_{A\subseteq \Theta }{\hat{m}}_{A,e(k)}}\;\;,\;\theta \subseteq \Theta ,\theta \neq \emptyset \end{array}\right.$$

Among them, k=3,⋯,L, $m_{\theta ,e(k)}$ reflects the degree of joint support for the proposition θ after the combination of the first k pieces of evidence, and t $m_{\theta ,e(1)}=m_{\theta ,1}$ and $m_{P(\Theta ),e(1)}=m_{P(\Theta ),1}$.

Assuming that the overall reliability of L independent evidence after combination is $r_{e(L)}$, the combination weight is $w_{e(L)}$, and $m_{P(\Theta ),e(L)}=\left(1-r_{e(L)}\right)/\left(1+w_{e(L)}-r_{e(L)}\right)$ can be obtained from Formula (20). Accordingly, $r_{e(L)}$ can be determined by the following formula:(22)$$r_{e(L)}=\left(1-m_{P(\Theta ),e(L)}(1+w_{e(L)})\right)/\left(1-m_{P(\Theta ),e(L)}\right)$$

The value of $w_{e(L)}$ should be between the maximum weight of L independent evidence $\max\{w_{i}\}$ and 1, so $r_{e(L)}(w_{e(L)})\in [r_{e(L)}(1),r_{e(L)}(\max\{w_{i}\})]$.

Through iteration and multi-layer ER [16], the final fusion evidence is obtained.
(23)$$e_{\left(L\right)}=\{(H_{n},\beta_{n,i}),\forall n=1,\cdots ,N;\sum_{n=1}^{N}\beta_{n,i}=1\}$$

On the basis of the evaluation results, the final results can be quantified according to the utility-based method. Assuming that the utility of the grade H is u(H), the performance state of the assessed object is calculated as follows:(24)$$U=\sum_{n=1}^{N}u(H_{n})\beta_{n}$$

### 3.4. Performance Evaluation Steps of Complex Equipment System

Based on the above conclusion, we can get the general steps of performance evaluation of complex equipment considering the resume information as shown in Figure 2:

Step 1: Build an equipment performance evaluation index system and obtain relevant index test information and resume information;

Step 2: Based on the index test information and the resume information, the index is converted into the belief distribution form of the evidence by using Formulas (3) and (4), respectively;

Step 3: Use Formula (7) or Formula (8) to calculate the weight of evidence;

Step 4: Use Formulas (12) and (13) to confirm the reliability of evidence;

Step 5: Use ER rules to perform hierarchical fusion according to the index system to obtain the performance evaluation result of the belief degree structure shown in Formula (23).

## 4. Case Study

Taking a certain inertial group as an example, the application of the proposed evaluation method in engineering practice is illustrated to verify the effectiveness of the proposed method.

The inertial group consists of a gyroscope and accelerometer and their accessories. The gyroscope measures the angular motion of the object, and the accelerometer measures the linear motion of the object [34]. In the error calibration test of the accelerometer and gyroscope, the zero-order coefficient, first-order coefficient and pulse quantity per unit time are directly output. Combined with the actual history of the IMU, and by taking the transportation mileage and maintenance as examples, the index system is established as shown in the Figure 3 [35,36]. There are six input indices at the bottom, so it is necessary to establish five grades and three layers for the evaluation model to realize the integration of the bottom indices to the middle level and then to the top level. For example, ER-1 indicates the evaluation process of the first level with the zero-order term and first-order term of the gyroscope as indices.

ER-1 and ER-2 together constitute the first stage; in the second stage, ER-3 gets the dynamic performance of IMU, and ER-4 gets the static performance; the third layer integrates the dynamic performance and static performance of IMU, which is the top level of the index system. The Figure 4 gives eight groups of data of the HT-G110604 laser IMU that was tested in the same environment for eight consecutive days.

The data of the zero-order coefficient and first-order coefficient ($D_{0x}$, $D_{0y}$, $D_{0z}$, $S_{gx}$, $S_{gy}$, $S_{gz}$) of the gyroscope and zero-order coefficient and first-order coefficient ($K_{0x}$, $K_{0y}$, $K_{0z}$, $S_{ax}$, $S_{ay}$, $S_{az}$) of the accelerometer are shown in the following figure.

After consulting the HT-G110604 Laser IMU Product Certificate, the current IMU is certified as “acceptable”. Resume information is as follows in Table 2.

### 4.1. Information Transformation Technique

In the engineering application, after the equipment is distributed, it is in a new product for a short time, and then it degenerates into a usable state. Similarly, after the equipment reaches the condition of being repaired, it will soon face retirement and reach the state of being scrapped, that is, the level is set to $\Theta =\{H_{1},H_{2},H_{3},H_{4}\}$, and it is new $H_{1}$, usable $H_{2}$, to be repaired $H_{3}$ and to be scrapped $H_{4}$.

The maintenance information input transformation model is selected as follows:(25)$$y_{1}=\frac{4.05-(0.045\times x_{1}+0.4\times x_{2}+x_{3})}{4.05}$$

The transportation mileage information input transformation model is selected as follows:(26)$$y_{2}=\frac{21-(x_{1}+10\times x_{2})}{21}$$

The reference values of each grade are set as follows in Table 3:

The belief distribution form of the input can be obtained by using Formula (3).

### 4.2. Weight and Reliability Calculation

(1) Weight calculation

In ER-1, the zero-order coefficient of the gyroscope is fused with the first-order coefficient to represent the influence of the gyroscope on the performance of IMU.

In ER-2, the zero-order term coefficient and the first-order term coefficient of the accelerometer are fused, and the influence of the accelerometer on the performance of IMU is discussed. In the error model, the zero-order term is often much more important than the first-order term. Combined with expert experience, the zero-order term coefficient weight is set to 0.8, and the first-order term coefficient weight is set to 0.2.

In ER-3, the fusion results of the gyroscope and accelerometer represent the dynamic performance of IMU, and the weights are all set to 1.

In ER-4, the weight of transportation mileage and maintenance is set to 1.

In ER-5, the dynamic performance reliability r of IMU is obtained by the coefficient of variation method, and the weight ω of static performance is obtained by normalization; then, the weight of static performance of IMU is 1−ω.

(2) Reliability calculation

In ER-1 and ER-2, the zero-order term and the first-order term of the gyroscope and the zero-order term and the first-order term of the accelerometer have the same information sources, which are all obtained by the coefficient of variation (CVBW) method.

In ER-3, the reliability of dynamic performance of IMU is obtained by integrating the reliability of the gyroscope and accelerometer with an evidential reasoning algorithm.

In ER-4, the reliability of transportation mileage $r_{T}=$ existing mileage/maximum mileage, and the reliability of maintenance information $r_{M}=$ storage time/full-service time.

In ER-5, the final inertial performance state is obtained by the fusion of static performance and dynamic performance of IMU. The reliability of static performance evidence is based on life, and the reliability of dynamic performance evidence is obtained by the fusion of ER-3. All are shown in Table 4.

### 4.3. IMU Performance State Evaluation

The belief distribution of the ER-1 gyroscope fusion results is shown in Figure 5. Most of its performance levels are between usable and to-be-repaired, and few are new products and to-be-scrapped; moreover, in eight tests, with an increase in time, the gyroscope performance shows a weak trend.

In the belief distribution of ER-2 accelerometer fusion results, as shown in Figure 6, the accelerometers are the same as gyroscopes, and their performance levels are mostly between usable and to-be-repaired, while those of new products and to-be-scrapped are few; moreover, in eight tests, with an increase in time, the performance of gyroscopes also shows an obvious trend.

In ER-3, the fusion result of the gyroscope and accelerometer is shown in Figure 7.

In the continuous test of gyroscope and accelerometer performance, the state is relatively stable without great change, but there are fluctuations, and the performance state gradually decreases with the increase of test times. This is consistent with the working mechanism and tactical indices of IMU. From the confidence distribution of usable and to-be-repaired grades in the figure, the confidence of the gyroscopes and accelerometers are concentrated in the middle, but the changing trends are different. The changing trend of gyroscopes is more gradual, while that of accelerometers is more prominent, which is related to the sensitivity of gyroscopes and accelerometers to working time. The fusion of the two will bring uncertainty to the fusion result. Therefore, the performance state of the IMU cannot be comprehensively or reliably evaluated only by the IMU test data, and its static performance needs to be investigated synchronously.

In ER-4, the two static performance indices of maintenance and transportation information in the resume are also converted according to the four reference grades of {$H_{1},H_{2},H_{3},H_{4}$}, and the input information of the indices is put in the same frame and merged, and the following results are obtained in Table 5.

It can be seen that, for the same IMU, when the dynamic test information and static resume data represent the IMU performance at the same time, the resume data shows that its performance is more inclined to “poor performance”, while the test data is more inclined to “good performance”. This is because in the process of maintenance, the test performance will be improved, and the test data will become better while the resume data will remain unchanged. For example, in complex systems, the process manufacturing and product designs cannot be perfect, and there are often some small defects, which will lead to frequent failures. Maintenance after each failure can “make the test data look good” and cannot cover up these defects. Similarly, these defects indicate that there is a problem in the performance state of a single machine, a piece of equipment or a system. Therefore, it is necessary to integrate the dynamic performance and static performance of IMU, so as to get a more accurate performance state.

In ER-5, by integrating the results of ER-3 and ER-4, the distribution results of performance state confidence of IMU considering resume information are as follows.

After the fusion, as shown in Figure 8, the results kept the previous trend and made the results clearer. In ER-1, the reliability is 0.83, and in ER-5, the reliability is 0.99, which greatly improves the reliability of the overall evaluation. On the one hand, it shows that the test information and resume information confirm each other; on the other hand, it shows that they complement each other, which makes the evaluation more complete and comprehensive.

This kind of situation belongs to the natural degradation of habitual group work, so it is only necessary to continue to maintain it according to the existing conditions.

### 4.4. Comparative Analysis

Three control groups were formed in the process of the case analysis. First, the comparison between dynamic evaluation ($C_{1}$) based on test data, which does not considere resume information, and static evaluation ($C_{2}$) based on resume information; the second is the comparison between the static evaluation ($C_{2}$) based on the resume information and the comprehensive performance evaluation ($C_{3}$) that considers resume information. The third is the comparison between dynamic evaluation ($C_{1}$) based on test data and comprehensive performance evaluation ($C_{3}$) that considers resume information.

This is convenient for research, and the results based on the scores are used for comparison.

It can be seen from the Figure 9 that the dynamic evaluation of the inertial group based on the test data produces the phenomenon of “false height”; this phenomenon may be caused by many factors, such as excessive maintenance, excellent test environment and so on. Furthermore, the phenomenon has been alleviated after the introduction of the resume information. It is worth noting that during the evaluation of the second time node, the red circle in the Figure 9, the introduction of resume information did not alleviate the phenomenon of “false height”. The evaluation value of $C_{3}$ is in the middle of evaluation value $C_{1}$ and evaluation value $C_{2}$, but more than evaluation value $C_{1}$ and less than evaluation value $C_{2}$ at the same time. This is because the reliability of the second test data fluctuates too much, which leads to the decrease of the evaluation value, and the reliability of resume information is improved. Therefore, the evaluation based on the test data does not produce the phenomenon of “false high” at this time. However, in the subsequent integration, the reliability remained low to maintain the accuracy of the evaluation. Therefore, in the evaluation of the second time node, the evaluation value $C_{3}$ was more than the evaluation value $C_{1}$ and less than the evaluation value $C_{2}$ at the same time.

In order to further verify the effectiveness of the method, this section uses BPNN (back propagation neural network) based on the quantitative data, AHP (analytic hierarchy process) based on the qualitative information and FR (fuzzy reasoning) based on the semi-quantitative information for comparison. In the experiment, eight groups of inertial combination data are the training group, and four groups of inertial group data are the test group. 

In the BPNN, the number of iterations is 50 times, and the maximum training number is $10^{4}$ times, and training requires the precision to remain at $10^{-6}$.

In AHP, gyroscope, accelerometer, transportation mileage and maintenance condition are selected as the criterion layer, and the scheme layer is $\left\{H_{1},H_{2},H_{3},H_{4}\right\}$, and the judgment matrix A is shown as
(27)$$A=\left[\begin{array}{cccc} 1 & 1 & 1.40 & 1.75\\ 1 & 1 & 1.40 & 1.75\\ 0.71 & 0.71 & 1 & 1.25\\ 0.57 & 0.57 & 0.8 & 1 \end{array}\right]$$

In FR, the fuzzy matrix R is set as: R = [2.3; 1.9; 10; 8; 3; 1].

Furthermore, the mean square error (MSE) and mean absolute error (MAE) are selected as evaluation precision indices.

The precision indices of various methods can be obtained by calculation, as shown in Table 6. It can be seen from Table 6 that the ER method has the highest accuracy, followed by FR, and the BPNN method has the lowest accuracy.

BPNN is the most basic neural network. Its output results are propagated forward and its errors are propagated backward [37]. It can fit any nonlinear function with multiple inputs and multiple outputs, so it can be applied in regression, clustering and other fields, and it has achieved good accuracy and effect [38]. However, in this paper, due to the lack of resume information, it is impossible to get a good training effect through data-driven methods. In addition, the black-box modeling method of BPNN is still uncertain, and it is not suitable for the performance evaluation of complex equipment considering the resume information [39]. The other improved algorithms based on neural networks also face the same problem. In the process of AHP evaluation, the decision-making scheme level $\left\{H_{1},H_{2},H_{3},H_{4}\right\}$ does not constitute a progressive relationship, which is inconsistent with reality, and the test dynamic data are difficult to use, so it has strong subjectivity. The accuracy of FR is slightly lower than that of the ER rule, and it also gets good accuracy. However, the process of FR transforming quantitative information into qualitative information will bring more uncertainty. The evaluation model based on ER rules has the advantages of traceability of reasoning process and interpretable results while maintaining high accuracy, so the proposed method can effectively evaluate the performance state of complex equipment systems.

## 5. Conclusions

Based on the characteristics of complex equipment, this paper analyzes the commonly used evaluation and information fusion methods and puts forward a performance evaluation method considering the resume information. Qualitative resume information is expressed quantitatively by the model, and ER rules are adopted to evaluate the equipment performance under the premise of considering the weight and reliability of test data and resume information indices. This method uses the first-order fitting coefficient of variation method and the average failure time-based reliability calculation method to calculate and process the data reliability for the test data and the resume information, respectively, and analyzes the index weight by using expert construction weight, thus constructing the general complex equipment performance evaluation index system and fusing the index information based on ER rules to obtain the evaluation results. This method considers the reliability of the acquired data and improves the accuracy of evaluation. By calculating the weight of the performance indices and distinguishing the importance of indices, the contribution of each indicator can truly be reflected in the evaluation results; ER rules can effectively integrate multiple pieces of information, deal with the uncertainty in the evaluation process, and finally get intuitive evaluation results. Finally, through a case study of an IMU, a performance evaluation case considering the resume information is given, and the effectiveness of the evaluation is verified.

Further research may include three aspects. First of all, Section 2.1 points out that the input information transformation is uniformly and linearly distributed and the method is relatively simple, so it is necessary to study the information transformation based on a nonlinear method. Secondly, it is necessary to study the influence of the asymmetry of grade transformation on the results. A multi-index evaluation, especially the performance evaluation with resume information, will lead to the inconsistency between the initial level and the final level of input information in practice, which needs further study. Finally, it is necessary to analyze the sensitivity of indices, test the reliability of expert weighting and the correlation of indices, and further optimize the evaluation model of complex equipment systems.

## Figures and Tables

**Figure 1 entropy-24-01811-f001:**
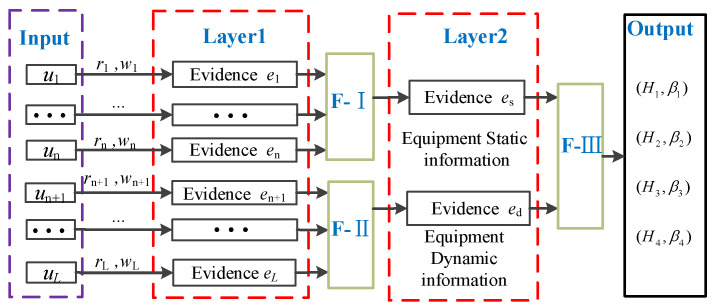
Performance evaluation system of complex equipment system.

**Figure 2 entropy-24-01811-f002:**
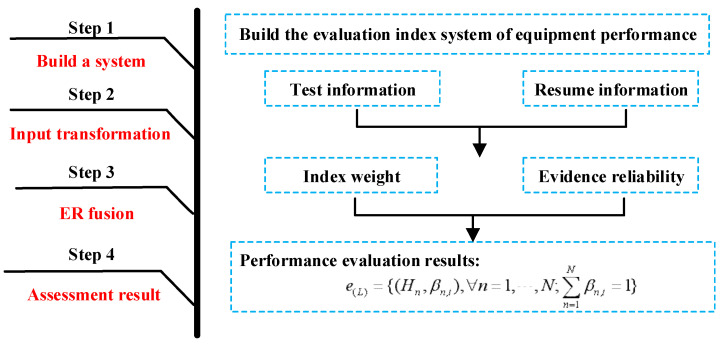
General steps of performance evaluation of complex equipment.

**Figure 3 entropy-24-01811-f003:**
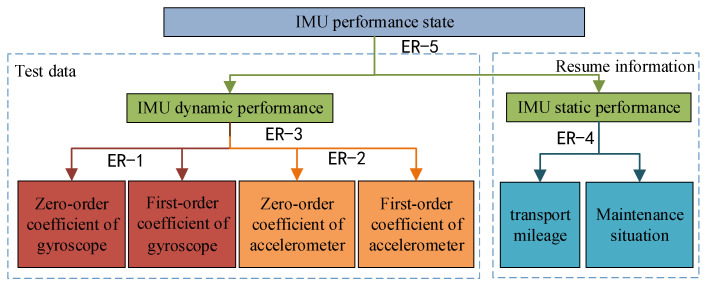
IMU performance evaluation system.

**Figure 4 entropy-24-01811-f004:**
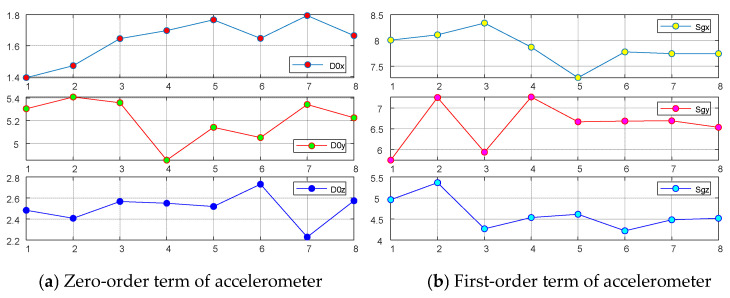
IMU test data.

**Figure 5 entropy-24-01811-f005:**
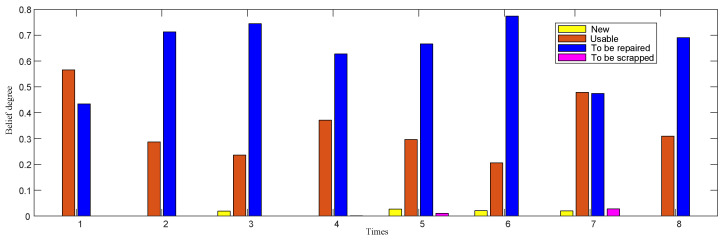
Belief distribution of gyroscope fusion results.

**Figure 6 entropy-24-01811-f006:**
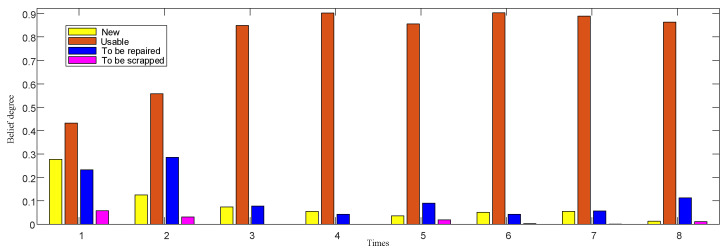
Belief distribution of accelerometer fusion results.

**Figure 7 entropy-24-01811-f007:**
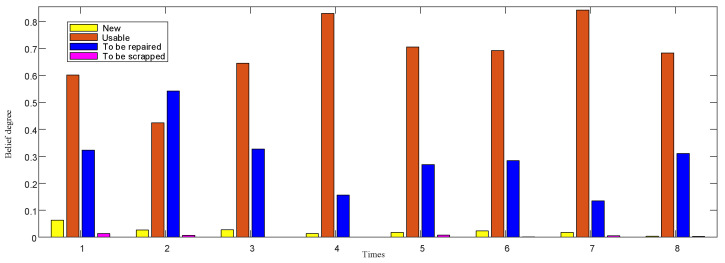
Belief distribution of IMU test data fusion.

**Figure 8 entropy-24-01811-f008:**
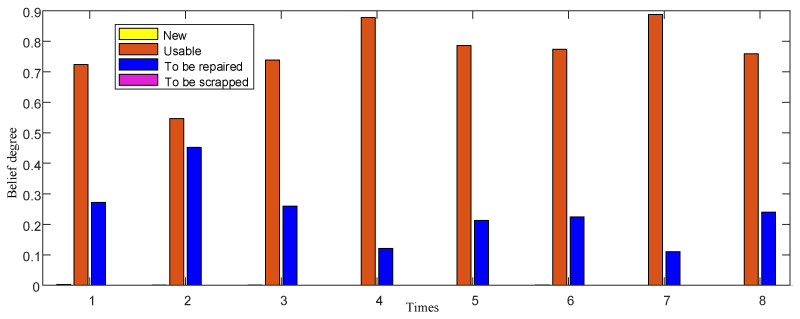
Belief distribution of IMU fusion.

**Figure 9 entropy-24-01811-f009:**
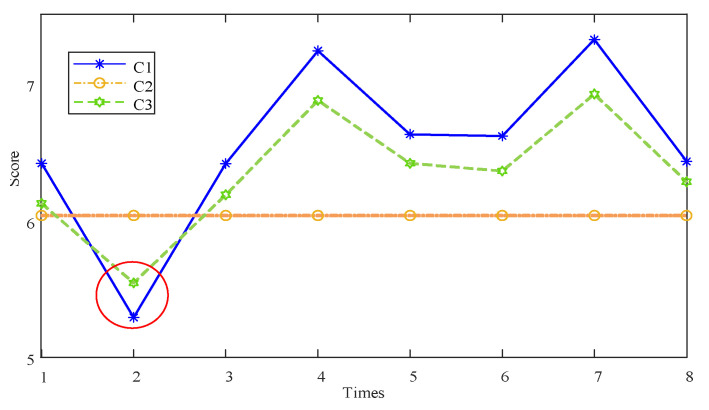
Score comparison chart.

**Table 1 entropy-24-01811-t001:** GJB 6288-2008 equivalent transformation coefficient of equipment maintenance.

Maintenance Level	Minor Repairs $\alpha_{1}$	Medium Repairs $\alpha_{2}$	Major Repairs $\alpha_{3}$
Equivalent coefficient	0.02–0.07	0.3–0.5	1

**Table 2 entropy-24-01811-t002:** Resume information.

Product Serial Number	Maintenance Information/Times			Transport Mileage /$10^{4}$ km		Enabled Duration/Hour	Effective Length of Service/Hour
	Minor Repair	Medium Repair	Major Repair	Highway	Railway		
1	4	2	1	0.5	0.7	120	5000

**Table 3 entropy-24-01811-t003:** Grade reference values.

Index	$K_{ax0}$ (g)	$K_{ay0}$ (g)	$K_{az0}$ (g)	$S_{ax}$ (g)	$S_{ay}$ (g)	$S_{az}$ (g)	$D_{0x}$ (deg/h)
$H_{1}$	14.0	21.6	19.90	8.0	1.30	2.0	1.00
$H_{2}$	14.3	21.7	19.98	8.3	1.35	2.1	1.25
$H_{3}$	14.7	21.9	20.02	8.8	1.45	2.2	1.75
$H_{4}$	15.0	22.0	20.05	9.0	1.50	2.3	2.00
Index	$D_{0y}$ (deg/h)	$D_{0z}$ (deg/h)	$S_{gx}$ (deg/h)	$S_{gy}$ (deg/h)	$S_{gz}$ (deg/h)	MI	TM
$H_{1}$	4.0	2.00	7.0	5.00	4.0	1.0	1.0
$H_{2}$	4.5	2.25	7.5	5.75	4.5	0.8	0.8
$H_{3}$	5.5	2.75	8.5	7.25	5.5	0.3	0.3
$H_{4}$	6.0	3.00	9.0	8.00	6.0	0.1	0.1

MI: Maintena nc, e MI:Information TM: Transport mileage.

**Table 4 entropy-24-01811-t004:** Weight and reliability.

ER-1	Index	Weight	Reliability	ER-2	Index	Weight	Reliability
	$D_{0x}$	0.8	CVBW		$K_{0x}$	0.8	CVBW
	$D_{0y}$	0.8			$K_{0y}$	0.8	
	$D_{0z}$	0.8			$K_{0z}$	0.8	
	$S_{gx}$	0.2			$S_{ax}$	0.2	
	$S_{gy}$	0.2			$S_{ay}$	0.2	
	$S_{gz}$	0.2			$S_{az}$	0.2	
ER-4	Index	Weight	Reliability	ER-5	Index	Weight	Reliability
	Maintenance information	1	$r_{T}$		Dynamic performance	ω	ER-3
	transport mileage	1	$r_{M}$		Static performance	1−ω	r(t)

**Table 5 entropy-24-01811-t005:** Static performance fusion results of IMU.

Index	$H_{1}$	$H_{2}$	$H_{3}$	$H_{4}$
Static performance of IMU	0	0.6079	0.3921	0

**Table 6 entropy-24-01811-t006:** Accuracy comparison of evaluation models.

Appraisal Procedure	ER	BPNN	AHP	FR
MSE	2.610	7.118	2.950	2.653
MAE	0.808	1.334	0.859	0.814

## Data Availability

Not applicable.

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
