# Peer review of "Performance Evaluation of Complex Equipment Considering Resume Information"

_entropy, 2022, doi:10.3390/e24121811_

Round 1
Reviewer 1 Report
This paper proposes a method to unify the test data and resume information and transform such information for evidential-reasoning-based performance evaluation of complex equipment. The proposed work aimed to address an important research question which is creating accurate and reliable approaches for evaluating the performance of complex equipment thus ensuring that the equipment is operating normally. Overall the paper is well written with good problem formulation and explanation of the methodology. However, there are some drawbacks that need to be considered to improve the quality of the manuscript.
The paper needs to clearly describe and define the term "complex equipment". The definition is vague and requires further clarification through more examples and use cases.
The order of citation numbers must appear in the paper based on their order of use in the manuscript (lines 70 and 71).
Provide examples of qualitative information and approaches to trace and measure qualitative information during the lifetime of complex equipment (line 81).
Authors need to explain and justify their methodological choice for information fusion (why evidential reasoning?) against other possibilities (such as subjective Bayes, fuzzy set theory, etc).
The introduction section must clearly explain the aim and objectives of the work. What are the challenges and why both quantitative and qualitative evaluation approaches should be considered?
What is the novelty of this paper? This needs to be clearly discussed in the paper.
The paper lacks to discuss related work. I suggest adding a section for Related Work discussion.
How can this work be compared against similar current state-of-the-art works?
The paper lacks to provide comparative evaluation/analysis/discussion results against current works in the literature.
Author Response
Based on the your comments , we have revised the paper very carefully. Thank you again for your review of the paper!

Reviewer 2 Report
The focus of this manuscript was focused on the evaluation method of complex equipment performance based on evidential reasoning (ER) considering resume information. A novel method is proposed to transform the test data and resume information in the ER-based performance evaluation. Overall, this manuscript’s writing and readability were sound quality. I do not have any comments.
Author Response
Thank you very much for your approval. The author will be encouraged to continue further research. At the same time, the author will infect the surrounding scholars and continue to create good works. Thank you very much!

Round 2
Reviewer 1 Report
Authors have applied all my comments and improved the quality of the manuscript.